# Flow Characteristics and Anti-Vortex in a Pump Station with Laterally Asymmetric Inflow

**Can Luo [1,*], Yufan He [1], Yinan Shang [2], Xiao Cong [3], Chao Ding [4], Li Cheng [1,*] and Shuaihao Lei [1]**

[1] College of Hydraulic Science and Engineering, Yangzhou University, Yangzhou 225009, China
[2] Jiujiang Hydraulic Waterpower Planning and Designing Research Institution, Jiujiang 332001, China
[3] Jiangsu Nantong Sanjian Construction Group Co., Ltd., Nantong 226100, China
[4] Jiangsu Hongyuan Bidding Acting Co., Ltd., Nanjing 210024, China
**\*** Correspondence: luocan@yzu.edu.cn (C.L.); chengli@yzu.edu.cn (L.C.)

**Abstract:** In a laterally asymmetric intake pumping station, the flow direction in the forebay is not consistent with flow in the intake channel. Thus, the adverse flow patterns, such as bias flow, large-scale vortex and asymmetric flow occur frequently in the forebay and sump. Based on the Reynolds-averaged Navier-Stokes (RANS) equation and the RNG k-ε turbulence model, a recent flow pattern in a laterally asymmetric intake pumping station was numerically simulated and analyzed, and effective vortex elimination measures were proposed. For the original scheme, seriously biased flow combined with large-scale vortices were observed in the forebay and several vortices occurred in the sump. To suppress the clash inflow in the south and north intake channel, the "straight diversion pier + curved wing wall" and "straight diversion pier + curved wing wall + V-shaped diversion pier" were installed separately. The" symmetrical 川-shaped diversion pier" and "symmetrical 川-shaped diversion pier + circular column" was utilized to eliminate the bias flow and large-scale vortices in the forebay. Finally, the "three- sectional diversion pier", "three- sectional diversion pier + triangle column" and "three- sectional diversion pier + triangle column + straight back baffle" was applied to decrease the vortex and asymmetric flow near the suction pipe of the sump. By attaching the rectification measure schemes in the intake channel and the forebay, the bias flow and large-scale vortex in the forebay were suppressed to varying degrees. The schemes significantly reduced the recirculation coefficient and greatly reduced the recirculation volume. By utilizing the vortex elimination measures in the sump, the vortex and asymmetric flow basically disappeared, the velocity distribution tended to become more uniform, and the flow rate distinction of each pump was smaller. The outcome can be used to provide a reference and basis for the improvement of flow pattern in similar laterally asymmetric intake pump stations.

**Keywords:** laterally asymmetric intake pumping station; vortex elimination; combined rectification; symmetrical 川-shaped diversion pier; recirculation coefficient

## 1. Introduction

According to the consistency of the flow direction between the intake channel and the forebay, the pumping stations are divided into front and lateral intake types. Due to the adverse flow pattern in the lateral intake pump, as shown in Figure 1b, the front intake type is privileged. However, recirculation may also appear in the front intake pumping station if design parameters such as the diffusion angle, are not reasonable, as shown in Figure 1a. The lateral intake forebay is also an absolutely essential option, owing to the limitations of the engineering layout. However, the lateral intake brings the recirculation, asymmetric flow and even vortices in the forebay due to the unsatisfactory inflow for the pump suction. Focused on the adverse flow pattern in the lateral intake forebay and sump, previous researchers have carried out a significant amount of research. Nasr, A. [1] compared and analyzed numerical simulations of the forebay in the lateral intake pumping

station forebay with, and without, rectification measures by using computational fluid dynamics (CFD) for multiple unit operations. Song, W. [2] used the method of numerical simulation and model testing to study the "Y" type diversion pier and "T" type diversion pier with the lateral intake pumping station as an example. Zhao, H. [3] used the measure of combining a guide-wall with a vertical column to improve the undesirable flow pattern of the water intake structure. Zhang, C [4] and Zhou, J. [5] used orthogonal experiments and computational fluid dynamics methods to analyze the flow characteristics of diversion piers with different combinations of parameters. Xi, W. [6] used an N-S equation and transport equation of turbulent dynamic power for the formation mechanism of the asymmetric adherent vortex in the side pump sump. Yang, F. [7] analyzed the flow field of a lateral intake pumping station and proposed a rectification scheme of a rectifier sill and diversion wall with openings. Luo, C. [8,9] simulated a multi-unit lateral inlet pumping station using a CFD by modeling triangular columns and partition piers to optimize the flow pattern. Choi, J. [10] used a combination of CFD and model experiments to study the vortices in the sump of a pumping station and its vortex elimination measures. Zhang, Y. [11], Zhou, J. [12] and Kadam, P. [13] used a combination of numerical simulation methods and physical models to research the poor flow patterns in forebay of the pumping station. Xia, C. [14] used square columns to conduct rectification research on the forebay of the pumping station. Chen, L. [15] simulated the steady flows in a typical rectangular sump based on the renormalization group analysis model and the SIMPLEC algorithm. Constantinescu, G. [16] used the standard k–ε equation to numerically simulate the vortex in forebay of a pumping station, and the simulated vortex structure in the pool was consistent with the results obtained from the model test. Xu, C. [17] simulated the flow patterns of the forebay based on FLUENT and proposed two types of rectification measures of diversion piers and pressure plates to improve the flow patterns. Ying, J. [18] proposed a combined rectification scheme with different spacing of columns and bottom stills for the poor flow patterns in the forebay of a large pumping station. Wu, X. [19] improved the flow patterns in the sump by adding flow rectification measures such as diversion piers and columns. Zi, D. [20] used numerical calculations and field experiments to study the effectiveness of combined diversion piers in improving the flow patterns in the intake structure of large pumping stations. Yu, Y. [21] conducted a numerical simulation and flow analysis on the inflow patterns of the diversion and intake pumping stations with side-inlets based on the realizable k-ε turbulent model and SIMPLEC algorithm, and found that the diversion grid had a significant effect on the regulation of the momentum distribution at the diversion section of the pumping station.

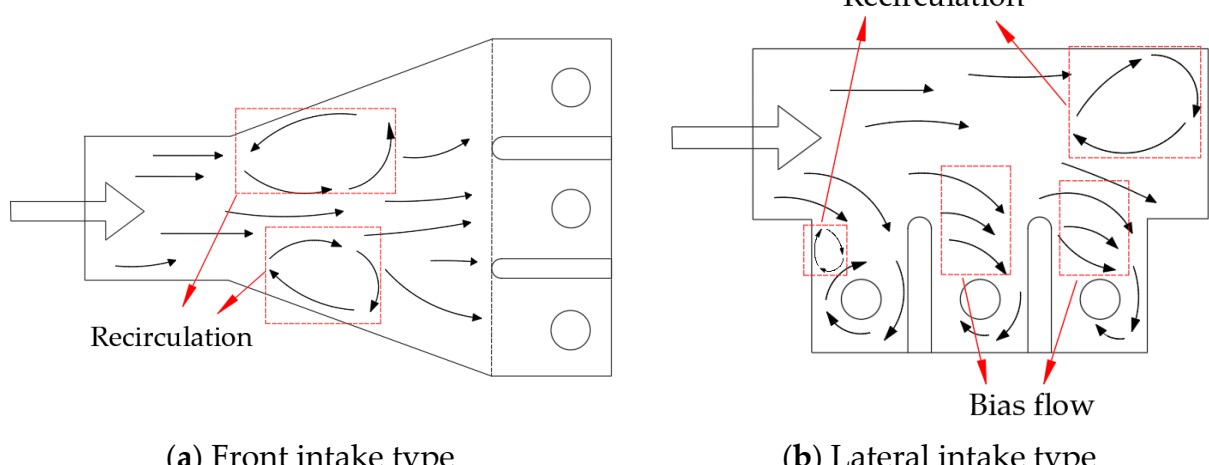

(**a**) Front intake type       (**b**) Lateral intake type

**Figure 1.** Flow pattern in the forebay and sump of front and lateral intake pumping station (ref arrows mean the flow direction).

In summary, many previous studies on flow patterns in lateral intake pumping stations have been carried out. However, research concerning pump stations with laterally asymmetric inflow are rarely reported. Recently, CFD method has grown into a mature technology, and is as important as experimental and the theoretical analyses. CFD has been used in the hydraulic machinery in pumping stations [22–27]. Therefore, this manuscript adopted a CFD method to analyze the recent flow pattern in a pump station with laterally asymmetric inflow and proposed the anti-vortex measures and rectification measures such as a V-shaped diversion pier and 川-shaped diversion pier. Moreover, the recirculation coefficient proposed by Zi, D. [28] was conducted to quantitatively analyze the range of large-scale recirculation zone and vortices. The outflow discharge on the outlet of the suction pipe was also discussed. The research not only weakened but solved the turbulent flow in the forebay and the sump, and provides an important reference for the improvement of flow pattern of other similar pumping stations.

## 2. Numerical Simulation

### 2.1. Computational Domains

The computational domain of a pump station with laterally asymmetric inflow was established by commercial interactive CAD/CAM software-Unigraphics NX 12.0, developed by Siemens A&D Groups (Berlin & Munich, German) [29]. The subdomains are the north intake channel (in green color), the south intake channel (in dark orange color), forebay (in purple color), sump (in gray color), and suction pipes (in gray color), as shown in Figure 2. The length of the south water channel is 16 D in the north-south direction and 6 D in the east-west direction. The length of the north water channel is the same as the south water channel in the north-south direction, but 4 D in the east-west direction. The length of the forebay is 23.25 D in the north-south direction and 18.25 D in the east-west direction. The length and width of the sump is 9.8 D and 3.4 D. Sudden starting of the pump will result into rapid water level drop in front of the pump bell mouth. If the water level is too shallow, the probability of surface vortex in the sump increases exponentially. Avoiding such conditions should be considered in which the reasonable design of the sump is a vital method. To avoid the surface vortex and other kinds of vortices, the Mandatory National Standard for pumping station design [30] is formulated for the designer of pumping stations in China. In the standard, the floor clearance and submergence depth of the bell mouth in the sump are advised as (0.6–0.8) D and (1.0–1.25) D. The submergence depth of the bell mouth was chosen as 0.8 D under the water surface. The distance from the back wall to the bell mouth was 1.5 D. Other parameter dimensions of the sump in this project were in accord with the standard, in which D is the diameter of the bell mouth inlet of the suction pipes.

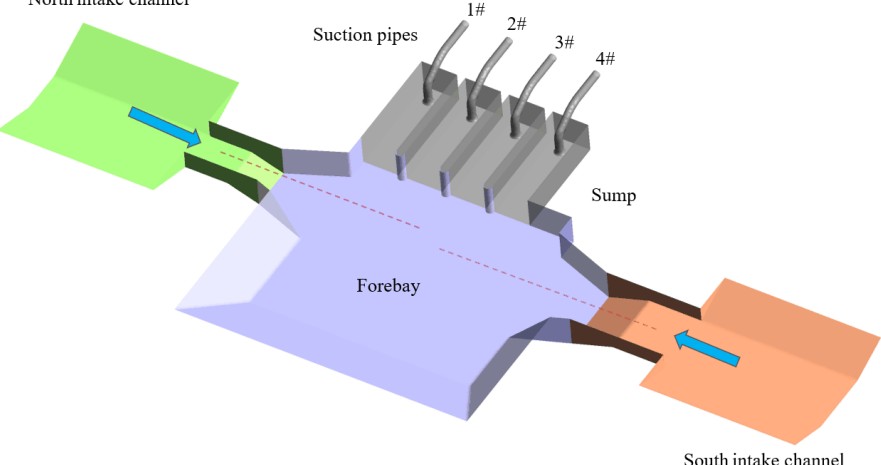

**Figure 2.** Computational domain.

## 2.2. Governing Equations

The laws of mass conservation, momentum conservation and energy conservation are the basic principles which the flow should follow. Considering no heat exchange in the energy conversion process of the forebay and the sump in the pumping station, the governing equations adopted in this study are the continuity equation and momentum equation. Due to the incompressible fluid, the continuity equation can be described as below:

$$\frac{\partial u_i}{\partial x_j} = 0 \tag{1}$$

The momentum equation is also called the N-S equation, it can be presented as:

$$\frac{\partial}{\partial t}(\rho u_i) + \frac{\partial}{\partial x_j}(\rho u_i u_j) = -\frac{\partial \rho}{\partial x_i} + \frac{\partial \tau}{\partial x_j} + \rho g_i + F_i \tag{2}$$

where $u_i$ is the three-dimensional velocity component of the fluid, $x_i$ is the three-dimensional coordinate component, $\rho$ is the pressure, $\tau$ is the stress tensor, $\rho g_i$ is the gravity term, and $F_i$ is the external source term. The stress tensor is as:

$$\tau = \left[ \mu_{ij} \left( \frac{\partial u_i}{\partial x_j} + \frac{\partial u_j}{\partial x_i} \right) \right] - \frac{2}{3} \mu \frac{\partial u_i}{\partial x_i} \tag{3}$$

where $\mu_{ij}$ is the dynamic viscosity and $\delta_{ij}$ is the Kronecker delta (when i = j, $\delta_{ij}$ = 1; when i $\neq$ j, $\delta_{ij}$ = 0).

## 2.3. Boundary Condition

The inlet of the intake channel was set as mass flow, which is the entrance of the computational domain, in which the discharge of the north and south intake channel is 3600 kg/s and 1770 kg/s. The exit of the suction pipe set as the static pressure is the outlet of the computational domain, and the reference pressure is 1 atm. The steady solutions are obtained by utilizing the finite volume method and mainly discussed in this research. The water surface is symmetry. The other surfaces, which are treated with scalable wall function, are no slip walls. Due to the subdomains contained in the computational domain, the interfaces between the intake channel and the forebay, the forebay and the sump, the sump and the suction pipe were set as the static interfaces. Reynolds number is a dimensionless number that can distinguish laminar flow from turbulent flow. In this simulation, the Reynolds number is much larger than 2300, thus the flow is fully developed turbulence. The turbulent model selection is greatly important. The RNG k-ε turbulence model uses the statistical technique of the renormalization group to correct the turbulent viscosity. This correction takes into account the swirling effect in the flow, which can better deal with the problem of large curvature flow. Therefore, considering the recirculation and vortex in the forebay and sump of the pumping station, the RNG k-ε turbulence model was utilized in this simulation and the convergence accuracy is $10^{-4}$. The SIMPLE algorithm and the second-order upwind scheme were used to simulate the flow pattern in the pump station.

## 2.4. Mesh Preparation and Independence Analysis

For the discretization of the computational domain, the quality and quantity of the mesh is critical. Therefore, a reasonable mesh generation strategy is very important for numerical simulations. Commonly, there are two types of mesh: structured and unstructured. Considering that the geometric characteristics and complexity of each subdomain were inconsistent, different mesh generation strategies were applied. The south intake channel, the north intake channel, the forebay and the sump adopted structured meshes. However, the suction pipes were considered as unstructured meshes because of their complex geometric profile. The mesh generation of each subdomain is listed in Figure 3. In

Figure 3e,f, the mesh around the diversion piers is encrypted, and the size of the mesh core is 0.1 m. In Figure 3f, the size of the densified grids closed to the triangle column is 0.08 m.

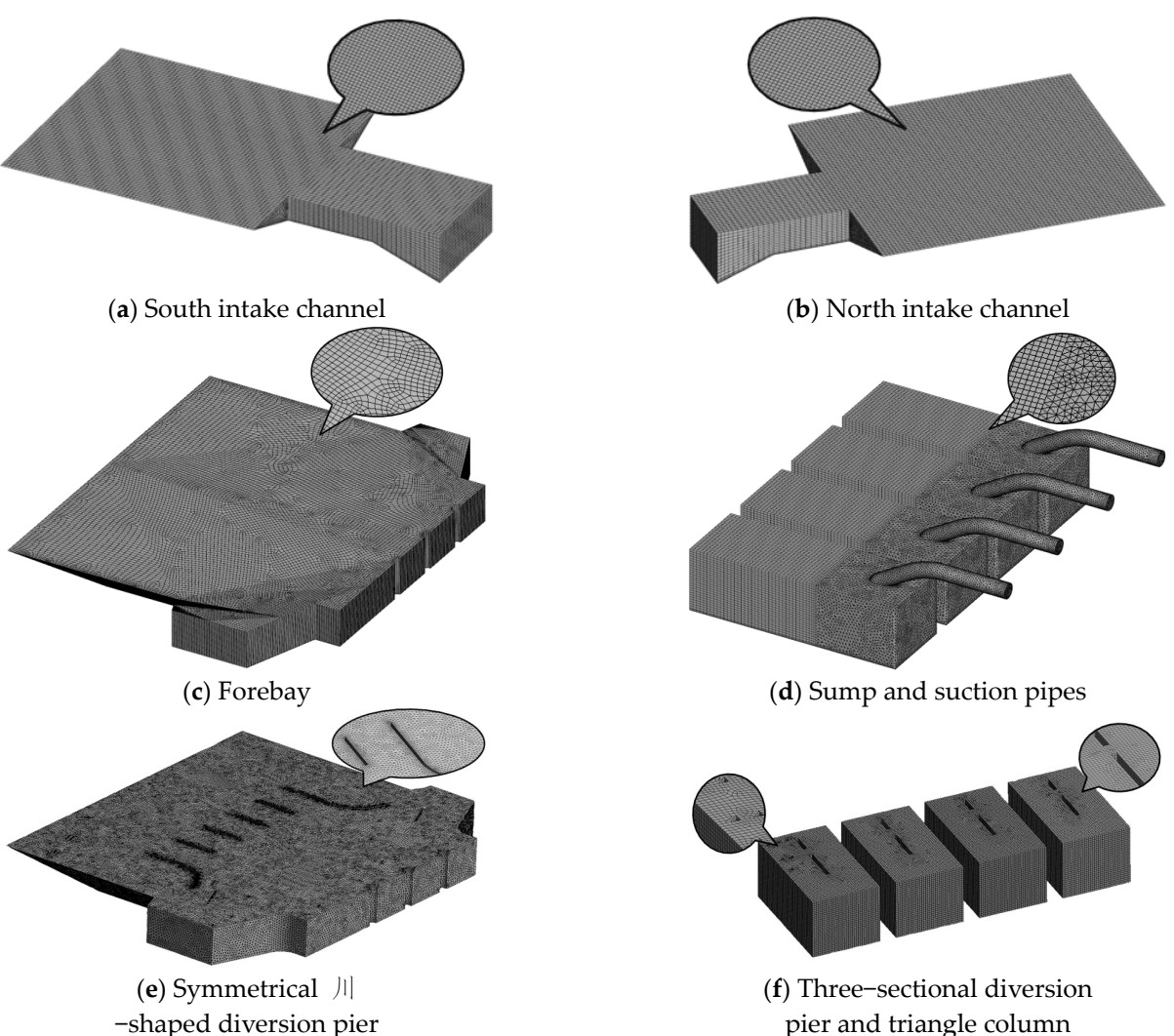

(**a**) South intake channel

(**b**) North intake channel

(**c**) Forebay

(**d**) Sump and suction pipes

(**e**) Symmetrical 川
−shaped diversion pier

(**f**) Three−sectional diversion
pier and triangle column

**Figure 3.** Mesh Preparation.

In general, more meshes are pursued to satisfy the accuracy requirements of calculation results. Nevertheless, the inappropriate mesh quantity also wastes computational resource and operation effectiveness. Thus, the determination of the mesh quantity should take into account both the stability of the calculation results and the minimization of the computational resources. Seven kinds of mesh quantities were generated: 0.48 million, 0.8 million, 1.02 million, 1.2 million, 1.86 million, 2.16 million, 2.86 million. The hydraulic loss $h$ was selected to analyze the grid sensitivity [31]. It can be presented as:

$$h = \frac{P_{\text{in}} - P_{\text{out}}}{\rho g} \qquad (4)$$

where $P_{\text{in}}$ and $P_{\text{out}}$ is the total pressure on the inlet and the outlet of the pump station, Pa; $\rho$ is water density, $1 \times 10^3$ kg/m$^3$; $g$ is gravitational acceleration, 9.8 m/s$^2$.

Figure 4 shows the trend of hydraulic loss for each scheme. The hydraulic loss decreases with the increasing mesh quantity. When the quantity was less than 1.2 million, the decreasing magnitude of the hydraulic loss was very obvious. With the further increase of meshes, the hydraulic loss difference of each scheme did not exceed 2%, which

indicates that the mesh quantity can satisfy the computational accuracy. Comprehensively considered, 1.2 million meshes was adopted to finish the following calculation.

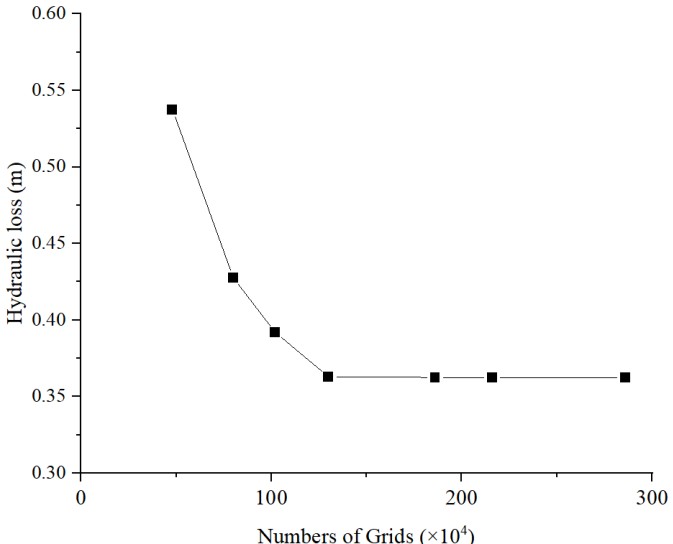

**Figure 4.** Hydraulic loss trend chart.

## 3. Analysis Sections and Parameters

### 3.1. Analysis Sections

To obtain and analyze the internal flow characteristics in the forebay and the sump, five sections are sliced. As shown in Figure 5, the horizontal sections are sliced as section 1, section 2 and section 3 respectively, which are used to analyze the flow patterns near the bottom, the middle and the surface. Moreover, the vertical sections $L_1$ and $L_2$ are sliced and marked as sections 4 and 5, observing the flow patterns at the inlet of the sump and before the suction pipe. The heights of sections 1–3 are 0.2 D, 1.5 D and 2.5 D above the bottom. The location of sections 4 and 5 are 0 D and 8 D away from the inlet of the sump.

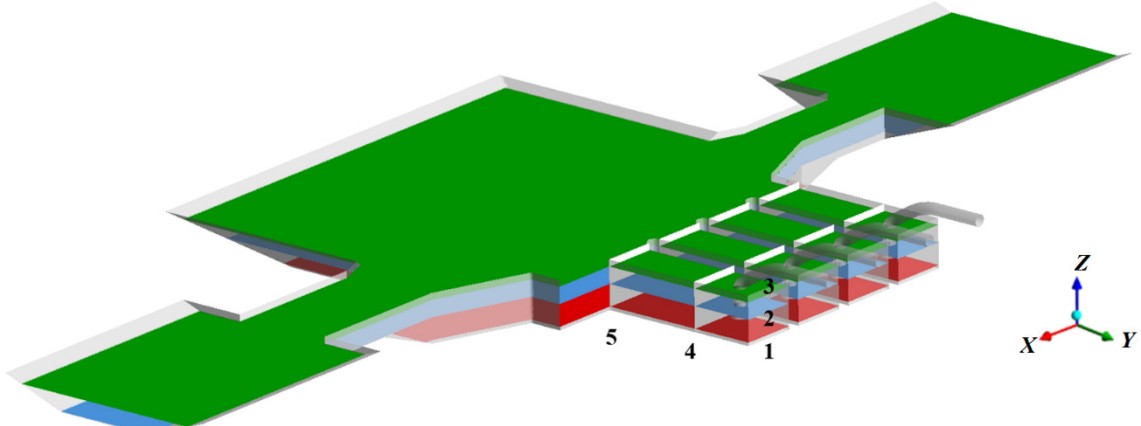

**Figure 5.** Analysis sections.

### 3.2. Analysis Parameters

The flow characteristics contain the velocity uniform feature and the vortex distribution performance. The axial velocity uniformity and axial velocity weighted average angle were utilized to quantitatively analyze the velocity uniform feature of the vertical sections 4 and 5 in the sump. Then the recirculation coefficient was proposed to quantitatively evaluate the vortex dimension in the forebay. More than that, the *Q* Criterion was also adopted to observe the vortex in the forebay.

(a)    Axial velocity uniformity and axial velocity weighted average angle [32].

The axial velocity uniformity $V_{au}$ reflects the uniformity of the axial velocity distribution on the overflowing section. When the axial velocity is more even, its value is closer to 1. Furthermore, the axial velocity weighted average angle $\theta_a$ is the angle between the flow direction and the overflowing section. When it flows more smoothly, its value is closer to 90°.

$$V_{au} = \left[ 1 - \frac{\sqrt{\sum_{i=1}^{n}(v_{ai}/v_a - 1)^2}}{n} \right] \times 100\% \tag{5}$$

$$\theta_a = \frac{\sum_{i=1}^{n}\left[\left(90° - \arctan\frac{v_{ti}}{v_{ai}}\right)v_{ai}\right]}{\sum_{i=1}^{n} v_{ai}} \tag{6}$$

where $v_a$ is the average axial velocity in m/s; n is the mesh node number; $v_{ai}$ is the axial velocity of each mesh node in m/s; $v_{ti}$ is the transversal velocity of each mesh node in m/s.

As shown in Figure 6, $v_{xi}$ and $v_{zi}$ is the velocity of each mesh node in X and Z direction. The transversal velocity $v_{ti}$ is the square root of the square sum of $v_{xi}$ and $v_{zi}$. $v_i$ is the velocity of each mesh node.

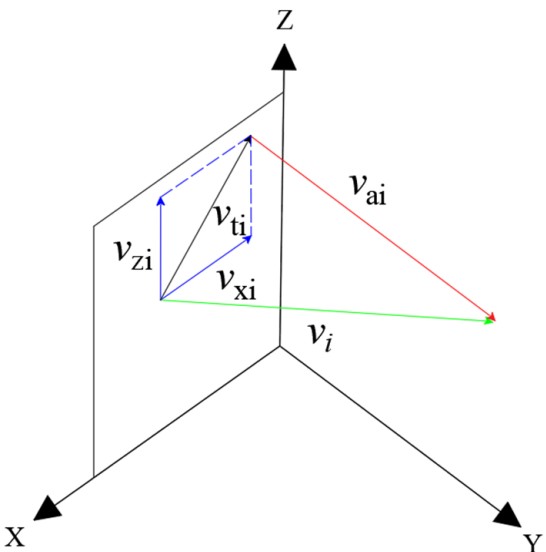

**Figure 6.** Velocity triangle on overflowing section.

(b) Recirculation coefficient

In order to study the influence of the vortex on the flow pattern in the forebay, the recirculation coefficient $C_h$ is proposed, which characterizes the range of the recirculation zone and vortex. The recirculation coefficient $C_h$ is the product of the recirculation volume ratio and the recirculation velocity ratio. The recirculation volume ratio equals to the value of recirculation volume divided by the volume. The recirculation velocity ratio equals to the value of average recirculation velocity divided by the average mainstream velocity. The equation of the recirculation coefficient $C_h$ can be conducted as:

$$C_h = \frac{Q_1}{Q_2} \frac{\left| \frac{\sum_{i=1}^{n} v_i}{n} \right|}{\frac{\sum_{j=1}^{m} v_j}{m}} \tag{7}$$

where $Q_1$ is the recirculation volume of each mesh element in the forebay and sump in m³; $Q_2$ is the volume of the forebay and sump in m³; $v_i$ is the velocity of each mesh element of the recirculation zone in m/s; $v_j$ is the mainstream velocity of each mesh element of the recirculation zone in m/s. $Q_1/Q_2$ is the recirculation volume ratio. $\left| \frac{\sum_{i=1}^{n} v_i}{n} \right| / \frac{\sum_{j=1}^{m} v_j}{m}$

is the recirculation velocity ratio. The value range of the recirculation coefficient is 0–1. For the ideal flow condition, there is no recirculation zone in the forebay and sump, and the value is 0. The flow pattern tends to more uniform as the decreasing recirculation coefficient. Therefore, the recirculation coefficient can capture the vortex and recirculation feature correctly.

(c) *Q* Criterion

The *Q* Criterion is based on the characteristic equation of the velocity gradient tensor. Hunt, J. [33] proposed to identify the region where the second matrix invariant *Q* is greater than zero as a vortex. The *Q* Criterion is:

$$Q = \frac{1}{2}\left(\|A\|_F^2 - \|S\|_F^2\right) \tag{8}$$

where *S* is the symmetric part of the velocity gradient tensor, and *A* is the antisymmetric part, respectively corresponding to the deformation and rotation in the flow pattern. Thus, the *Q* Criterion represents the physical significance of the vorticity over that of the deformation that dominates the rotating part of the flow pattern.

## 4. Anti-Vortex and Rectification Schemes

Combined anti-vortex schemes in the forebay listed in Table 1 were proposed to improve the flow pattern and weaken the large-scale recirculation zone in the forebay. For the "straight diversion pier + curved wing wall" scheme, it is marked as scheme 1, straight diversion piers of 5 D and 7 D length were installed in the south and north intake channels, respectively, the straight wing wall before the sump was replaced with a curved wing wall with a radius of 1.5 D. For the "straight diversion pier + curved wing wall + V-shaped diversion pier" scheme marked as scheme 2, the V-shaped diversion pier of 2 D length and 0.2 D width was put forward on the inlet of the forebay. The angle of the V-shaped diversion piers was 60° to avoid the clash inflow. For the "symmetrical ノ||-shaped diversion pier" scheme marked as scheme 3, the symmetrical ノ||-shaped diversion pier was set in the forebay for a smoother flow and to guide the flow into the sump. For the "symmetrical ノ||-shaped diversion pier + circular column" scheme marked as scheme 4, two groups of circular columns were set near the symmetrical ノ||-shaped diversion pier and before the inlet of the sump to eliminate the vortex and improve the axial velocity uniformity. In Table 1, new additional anti-vortex measures for each scheme are distinguished with an orange color.

To solve the vortices in the sump and asymmetrical inflow before the suction pipes, some combined rectification schemes of "three- sectional diversion pier", "three- sectional diversion pier + triangle column" and "three- sectional diversion pier + triangle column + straight baffle plate" were proposed to be installed in the sump, as shown in Table 2. For the "three-sectional diversion pier", it is marked as scheme 52, straight three-sectional diversion piers with a height of 4 D were installed in sump 1#–3# and "J"-shaped three-sectional diversion pier in sump 4 was installed in sump 4#. The distance between each diversion pier was 0.2 D. For the "three- sectional diversion pier + triangle column" scheme marked as scheme 6 in Table 2, two groups of triangle columns with a length of 0.2 D were arranged in sump 4#. For the same group of triangle column, the distance between the neighbouring triangle column was 0.25 D. However, the distance between different groups of triangle columns was 1.5 D. For the "three- sectional diversion pier + triangle column + straight back baffle" scheme marked as scheme 7 in Table 2, the straight back baffle was applied at the back wall of the sump to eliminate the tiny vortex and the asymmetrical back flow. Its length, width and height were 1.5 D, 0.4 D and 2.8 D, respectively. In Table 2, the rectification measures for each scheme are distinguished with a red color.

**Table 1.** Anti-vortex measures in the forebay.

| Scheme | Anti-Vortex Measure | Dimension |
|--------|--------------------|-----------|
| — | Original Scheme | |
| 1 | Straight diversion piers and curved wing wall | |
| 2 | V-shaped diversion pier | |
| 3 | symmetrical 川-shaped diversion pier | |
| 4 | Circular Column | |

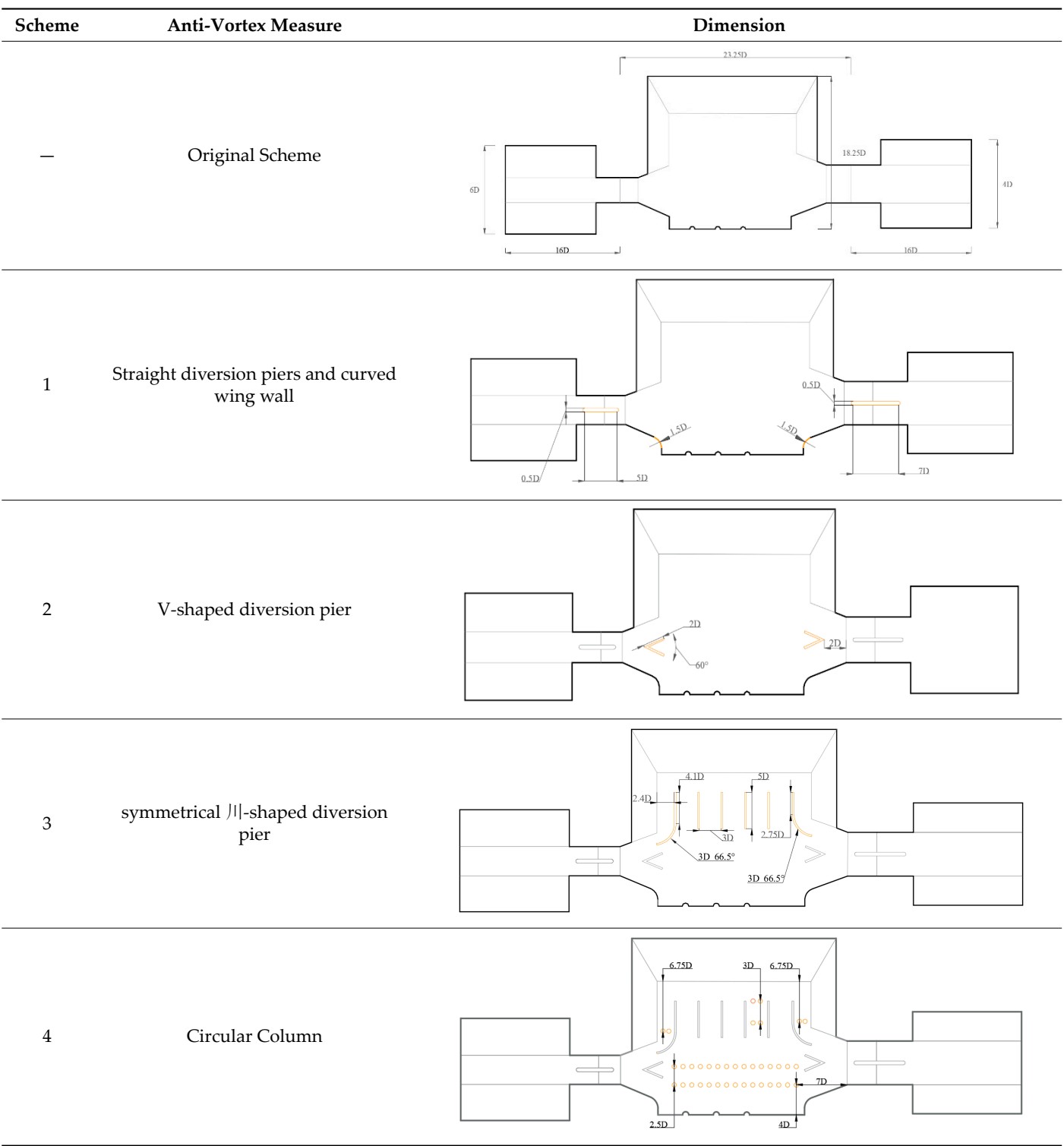

**Table 2.** Rectification measures in the sump.

| Scheme | Rectification Measure | Dimension | |
| --- | --- | --- | --- |
| — | Original scheme | 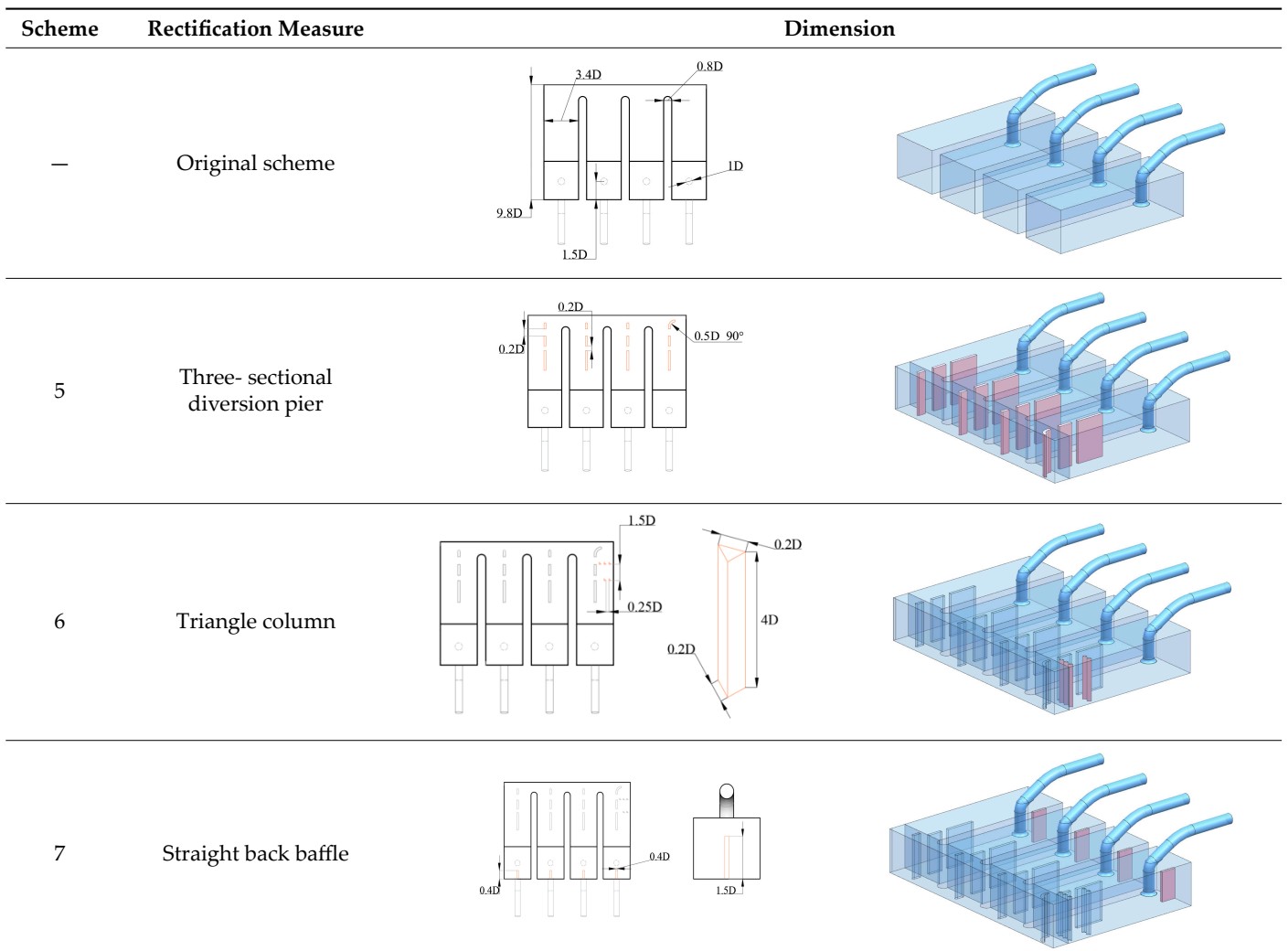 | |
| 5 | Three- sectional diversion pier | | |
| 6 | Triangle column | | |
| 7 | Straight back baffle | | |

## 5. Result Analysis

### 5.1. Anti-Vortex Effect in the Forebay

#### 5.1.1. Flow Pattern Improvement

Figure 7 is the surface (Section 3) streamline and axial velocity contour chart of each scheme. The red arrows mean the flow direction. In Figure 7a, the flow pattern of the original scheme in the forebay and sump is extremely disorganized. Two recirculation zones are located in the north and south intake channels recorded as NRZ and SRZ. Moreover, the SRZ is much larger than NRZ due to the discharge of north intake channel, which is double that of the south intake channel. The formation of NRZ and SRZ contributes to the clash flow caused by the huge discharge difference. Another large-scale recirculation zone nearly covering the entire forebay is recorded as FRZ. Moreover, the offset flow (OF) occurs in all sumps, and the circumfluence flow (CF) exists in the sump 3# and 4#. The vortex is observed in the sump 4#, recorded as SV. In Figure 7b, the flow pattern in the forebay is slightly improved. The NRZ disappears and is absorbed to breed a new FRZ. The SRZ is pushed away from the sump. The dimension of circulation zones obviously reduces. The range of CF tends to be lower but some SVs are born in sump 3# and 4#. In Figure 7c, the distribution of the recirculation zones differs significantly. The SRZ eliminates due to the V-shaped diversion pier, but a FRZ is refreshed in the eastern forebay away from the sump. The former FRZ is forced to move towards to be away from the sump and then is broken up into two, and the NRZ reforms. The mainstream is split to two sub-streams. The western

sub-stream of north intake channel is guided in sump 3#–4# and the western sub-stream of south intake channel is guided in sump 1#. The eastern sub-stream of north intake channel flows in sump 1#–2#. Yet, the eastern sub-stream of south intake channel visits the forebay and finally circles into sump 1#. The vortices behind the V-shaped diversion pier installed in the forebay are born, which are recorded as FVs. By the way, the CFs and SVs still exists. In Figure 7d, The FRZ is well split with dimension reduction when the symmetrical 川-shaped diversion pier is adopted. However, some tiny vortices or recirculation zones are distributed instead in the forebay. The SRZ and NRZ are surveyed. The OFs in sump 1#–2# eliminate. The OFs and CFs in the other sumps are improved, especially in sump 3#. The range of SV in sump 3# decreases obviously. In Figure 7e, the circular column is helpful to suppress the tiny vortex and recirculation zone in the forebay, making the stream in the forebay smoothly. The FRZ, NRZ and SRZ are locked in the labyrinthine created by the combined anti-vortex measures, benefiting the flow pattern in the forebay and sump. In sump 1#–3#, no CF, SV and OF is observed. In sump 4#, the CF and SV still exist. Nevertheless, the asymmetrical inflow of the suction pipe should be noted for every scheme.

　　Figure 8 is the middle (Section 2) streamline and axial velocity contour chart of each scheme. Overall, the flow pattern in the middle is similar with the surface. In Figure 8a, the FRZ, SRZ and some small vortices are observed in the forebay. The range of the FRZ extends to the location of the NRZ on the surface. The OFs also happen in all sumps. The CVs and vortices are seen in sump 3# and 4#. The size of the vortices is more obvious than it on the surface. The stream from the south intake channel is mainly flows into sump 1#, and the stream in the other sumps is originated from the north intake channel whose main-stream is badly squeezed. In Figure 8b, the range of the FRZ decreases, but the number of the FRZ increases. The SRZ is forced to the direction away from the sump and its dimension expands. The stream from the south intake channel flows into sump 1# and 2#. On the opposite, the stream from the north intake channel flows into sump 2#–4#. The inflow squeeze state is improved. The OFs, SVs and CFs of each sump are weakened. In Figure 8c, the flow patterns in the forebay such as the SRZ, FRZs and FVs are similar to those on the surface. But the streamline in the sump tends to deteriorate more than that on the surface. In Figure 8d, the SRZ and NRZ are forced to the forebay. The FRZ is improved and its dimension decreases. The smoothness in the sump 1# and 2# is ameliorated. However, the flow pattern in the sump 3# and 4# is basically the same as Scheme 2. In Figure 8e, there are still recirculation zones in the forebay, but their dimension decreases. The flow pattern in the sump is similar with Scheme 3. The inflow in sump 1#–3# is smooth. For all sumps, the stream near the suction pipe is unsatisfactory, especially the asymmetrical flow and SVs. Therefore, the installation of rectification measures in the sump is essential.

　　Figure 9 is the bottom (Section 1) streamline and axial velocity contour chart of each scheme. In Figure 9a, the flow pattern at the bottom is similar with Sections 2 and 3. Among the three horizontal sections, the inflow of the north intake channel is mostly squeezed. In other words, the range of recirculation zone is largest. There are three vortex cores in the forebay. Due to the influence of clash flow, large scale recirculation zone (SRZ) is more likely to be generated near the inlet with small discharge. Another recirculation zone (FRZ) is located in the forebay. The flow pattern in the sump for each horizontal section is similar, but the strength of the SV is maximum. In Figure 9b, the squeezed inflow is well improved. The range of recirculation zone decreases. However, the OFs, CFs and SVs in the sump are not significantly improved. In Figure 9c, the flow pattern at the bottom is also similar to Sections 2 and 3. The improvement of the flow pattern in the forebay is not obvious. Moreover, the OFs, CFs and SVs in the sump are growing, and are not suppressed. In Figure 9d, the range of FRZ diminishes, yet the range of SRZ and NRZ expands. The OFs, CFs and SVs in the sump are still unsatisfactory. In Figure 9e, large-scale recirculation zones basically disappear. Nevertheless, the SVs are widespread in all sumps.

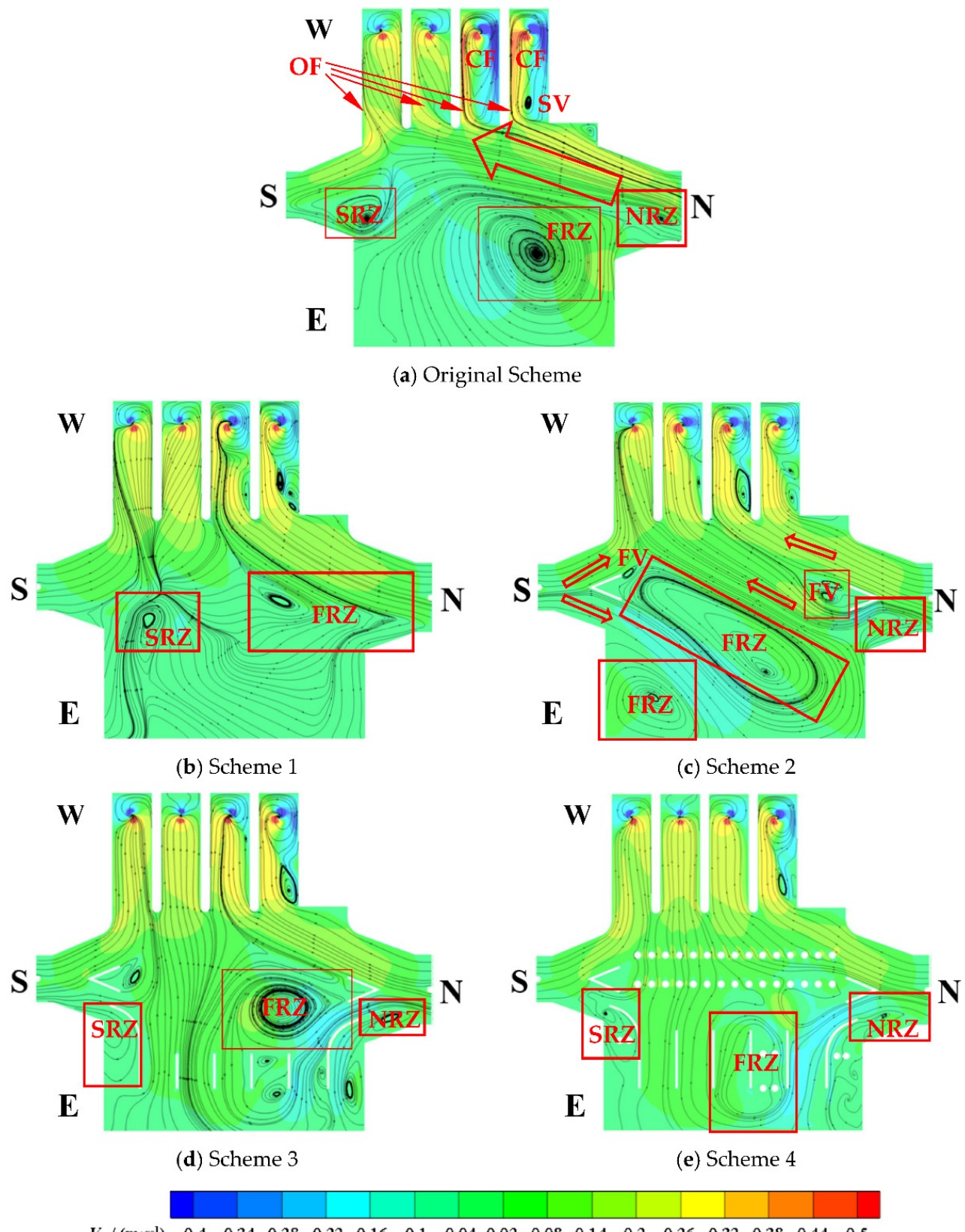

**Figure 7.** Surface streamline and axial velocity contour chart.

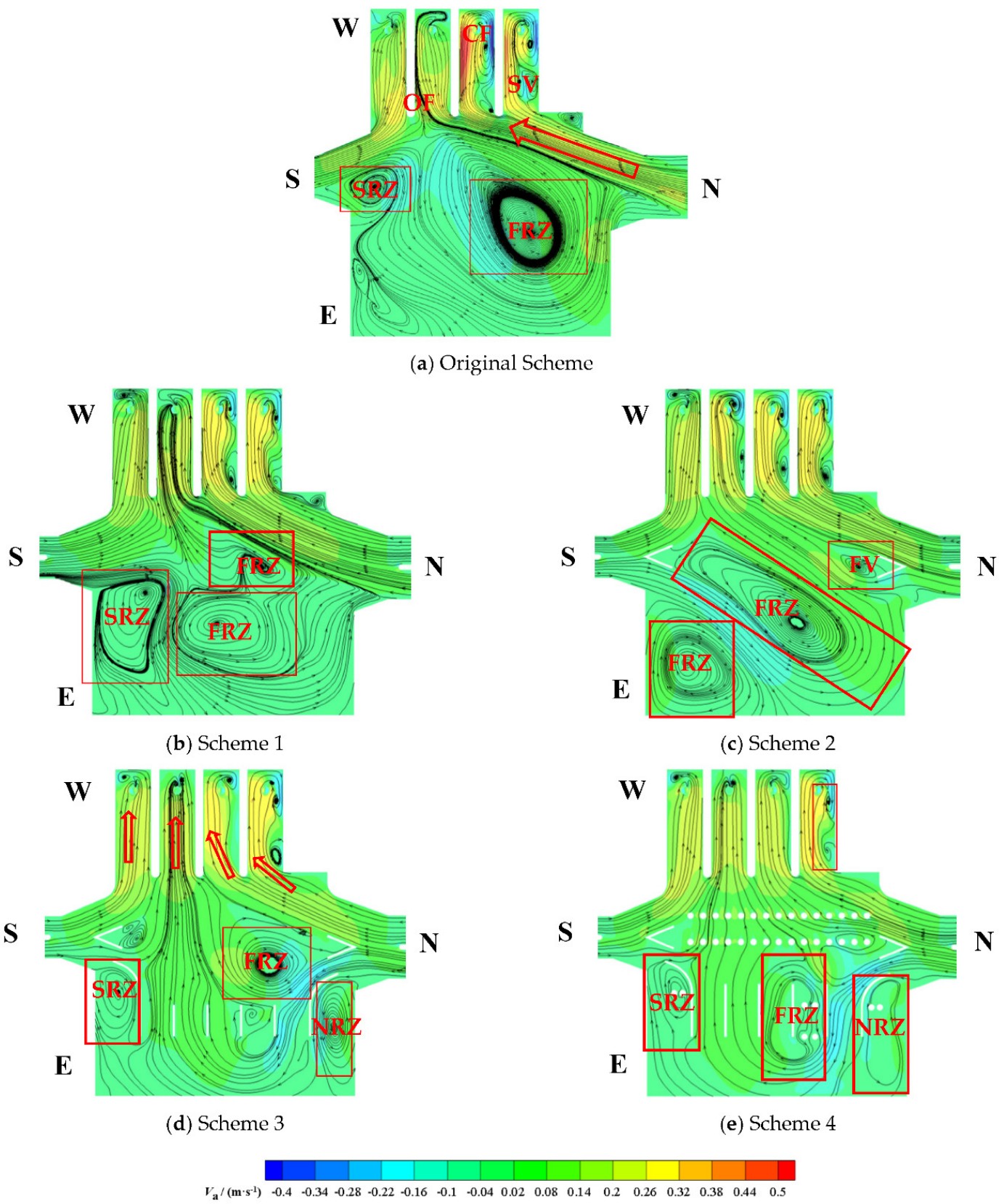

**Figure 8.** Middle streamline and axial velocity contour chart.

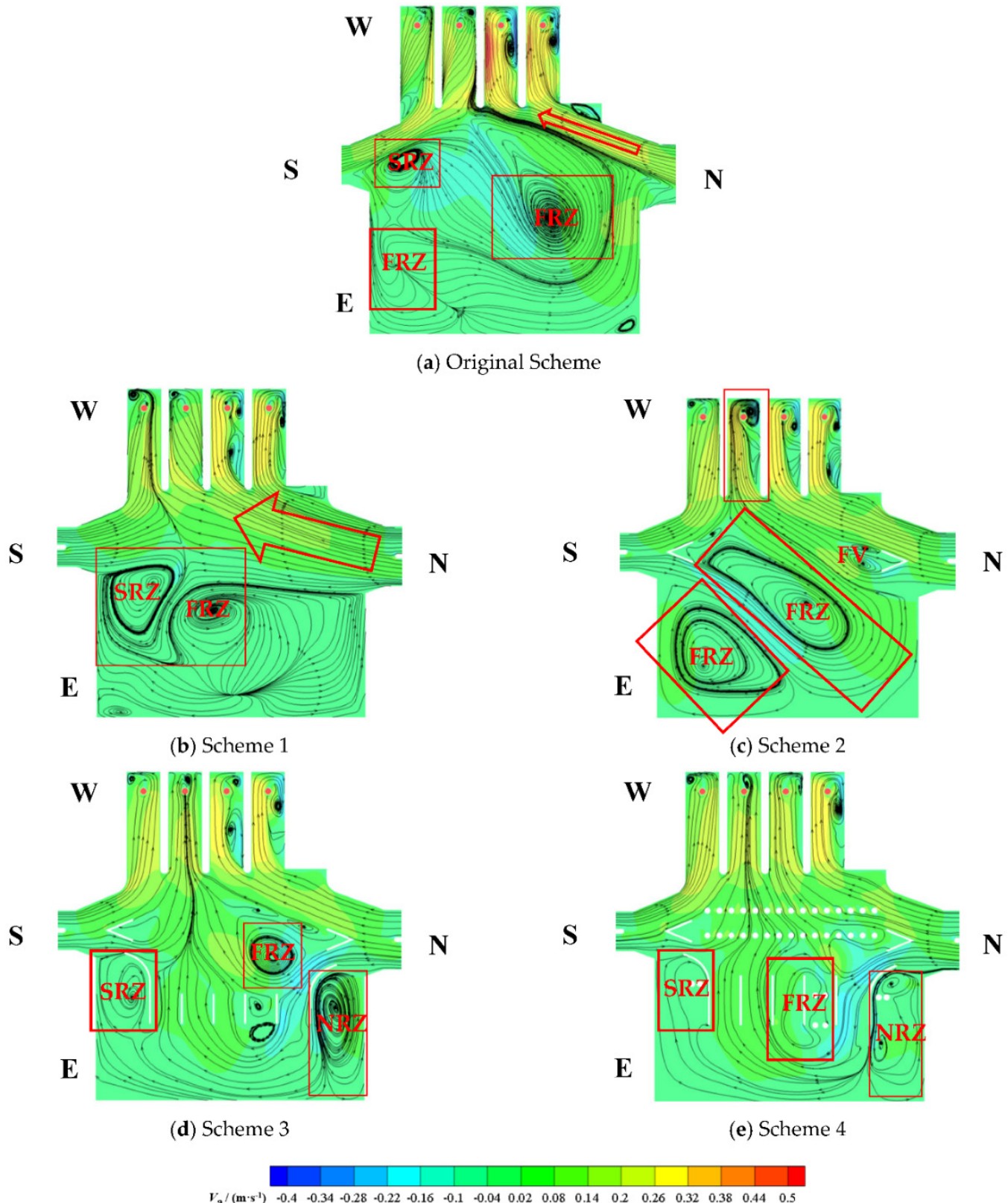

**Figure 9.** Bottom streamline and axial velocity contour chart.

### 5.1.2. Vortex Distribution

Figure 10 is the vortex distribution contour in the forebay by utilizing *Q* Criterion. For the original scheme, vortices frequently occur, spreading over nearly 40% of the forebay. As shown in Figure 10b–e, the vortex is eliminated in the intake channel by the straight diversion pier and curved wing wall for scheme 1. The vortex at the entrance of the forebay is significantly reduced, resulting in the initial improvement of the flow pattern. However, there are still large vortices and backflow at the wing wall and the center of the forebay. For scheme 2, V-shaped diversion piers at the inlet of the forebay is arranged to guide the stream and improve the flow pattern. The vortex near the wing wall is solved. Meanwhile, a new vortex occurs behind the V-shaped diversion pier. More than this, two large scale recirculation zones are observed in the forebay. For scheme 3, the flow pattern is further smoothed with the symmetrical 丿||-shaped diversion pier. The large-scale recirculation zone is eliminated and replaced by some tiny vortices. For scheme 4, the circular columns are utilized to remove the vortices in the forebay. The dimension of the vortices decreases rapidly.

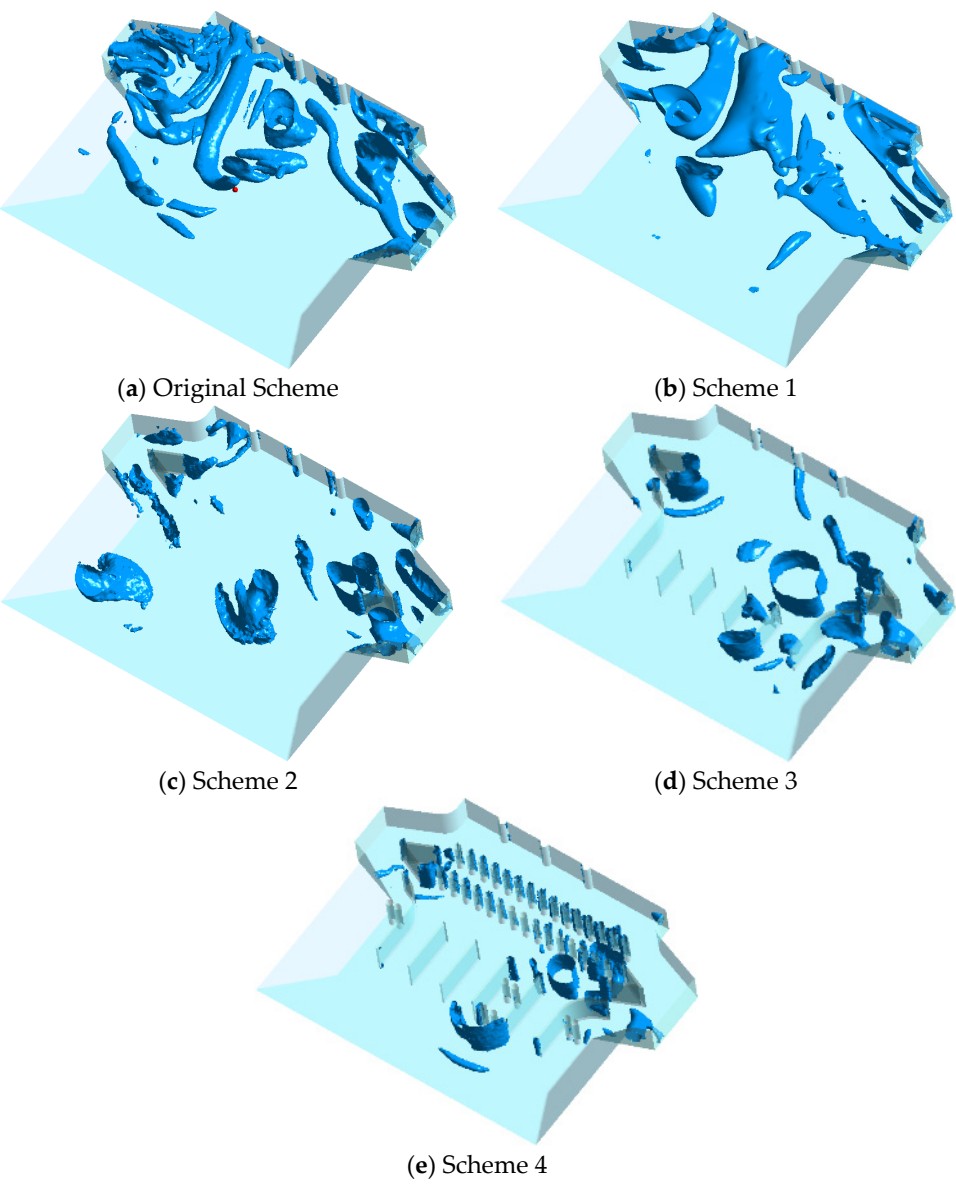

(**a**) Original Scheme            (**b**) Scheme 1

(**c**) Scheme 2            (**d**) Scheme 3

(**e**) Scheme 4

**Figure 10.** Vortex distribution in forebay.

The recirculation coefficients of each scheme are calculated using Equation (7), which are listed in Table 3. It can be seen that the recirculation coefficient of the original scheme is 0.347, and the recirculation coefficients of each scheme are 0.233, 0.206, 0.202 and 0.193 respectively, meaning the recirculation zone and vortex are well treated. For scheme 4, the range of recirculation zone and vortex is reduced by 44.4%.

**Table 3.** Recirculation coefficient of each scheme.

|  | Original Scheme | Scheme 1 | Scheme 2 | Scheme 3 | Scheme 4 |
|---|---|---|---|---|---|
| Recirculation volume ratio | 0.396 | 0.261 | 0.352 | 0.263 | 0.263 |
| Recirculation velocity ratio | 0.875 | 0.893 | 0.583 | 0.766 | 0.735 |
| Recirculation coefficient | 0.347 | 0.233 | 0.206 | 0.202 | 0.193 |

### 5.1.3. Hydraulic Loss Performance

The hydraulic loss performance is highly important for the pump station operation. Hence, hydraulic losses data of different schemes are calculated by Equation (4) and drawn in Figure 11.

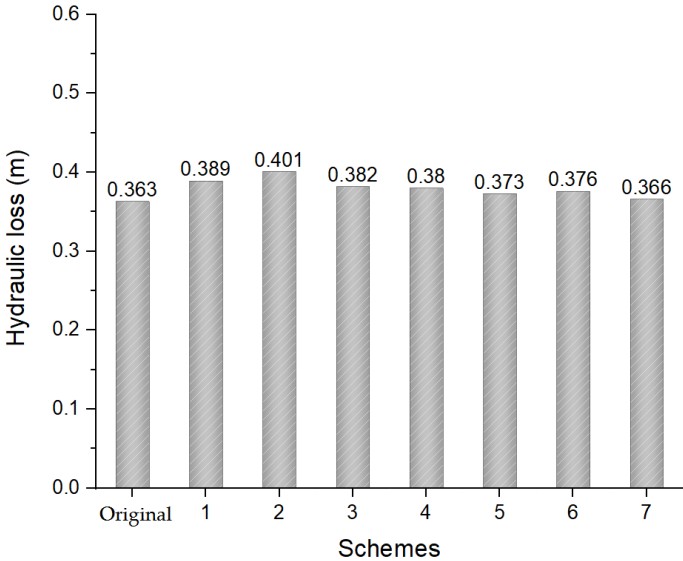

**Figure 11.** Hydraulic loss trend chart of each scheme.

For original scheme, the hydraulic loss is 0.363 m. The installation of anti-vortex measures in the forebay and sump results in the increase of hydraulic performance. Among the anti-vortex schemes, the hydraulic loss of scheme 2 is maximum, and its value is 0.401 m, increases by 10.46%. The hydraulic loss of scheme 7 is minimum, and its value is 0.366 m, which is almost equal to the original scheme. Usually, large-scale recirculation and vortices consume vast energy. Therefore, the hydraulic loss of original scheme mainly contributes to the adverse flow pattern. Although the attachment of each anti-vortex measure is beneficial for the flow pattern and suppress the vortices, the local hydraulic loss increases sharply. Hence, the hydraulic loss of the anti-vortex measure scheme is due to the recirculation and vortices, but mainly depends on the local hydraulic loss.

### 5.2. Rectification Efficiency in Sump

#### 5.2.1. Axial Velocity Uniformity

To gain the vertical axial velocity distribution on the inlet of the sump and near the bell mouth, the axial velocities on the characteristic lines of the Sections 4 and 5 are extracted, and the characteristic lines are marked as line 1–6, shown in Figure 12.

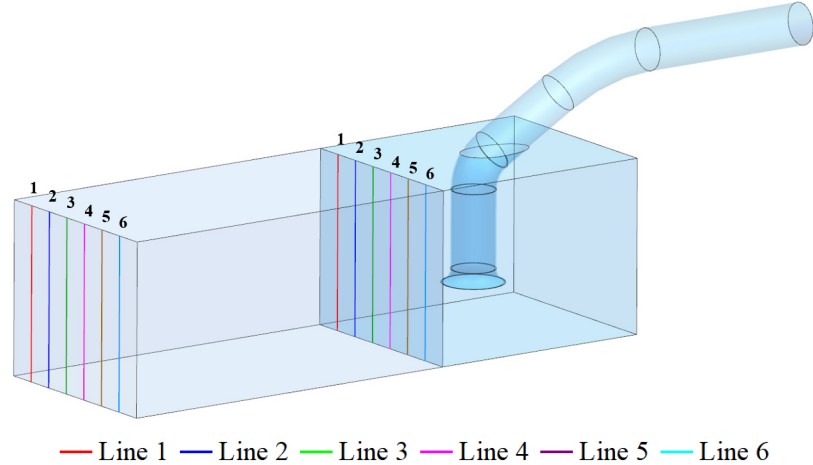

**Figure 12.** Diagram of characteristic lines on the inlet of the sump and near the bell mouth.

Figures 13–16 are the axial velocity chart on the inlet of the sump. In which, "#" is the number symbol of the unit. In Figure 13, a huge axial velocity difference in sump 1# from the bottom to the surface is captured for the original scheme. When the height is 0.6 m, the axial velocity is obviously smaller than the other heights. For rectification schemes, such situation is well improved. In Figures 14 and 15, the axial velocity distribution of original scheme on the inlet of sump 2# and 3# differs too much. The axial velocity increases from the bottom to the surface. By applying the rectification measures, the axial velocity decreases and becomes more uniform. As shown in Figure 16, the axial velocity is uneven. After arranging the rectification measures, the improvement is not obvious.

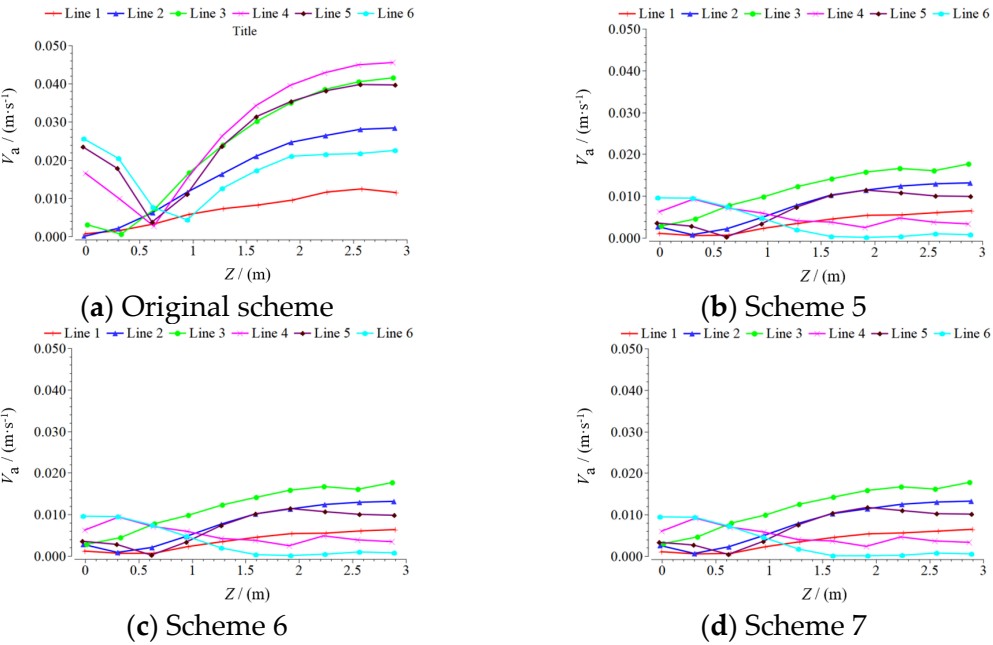

**Figure 13.** Diagram of axial velocity on the inlet of the sump 1#.

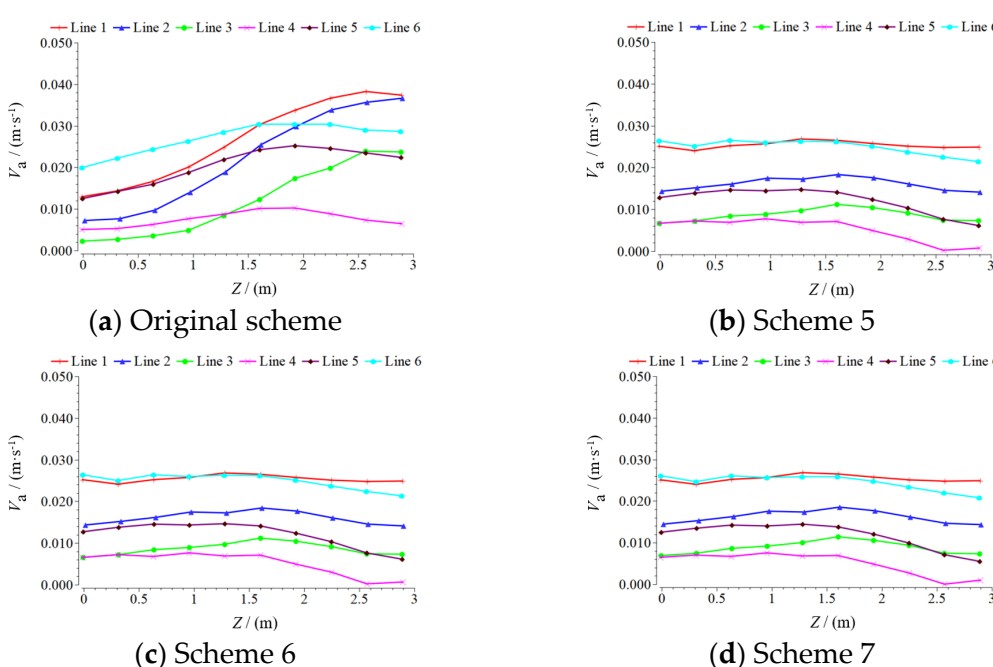

**Figure 14.** Diagram of axial velocity on the inlet of the sump 2#.

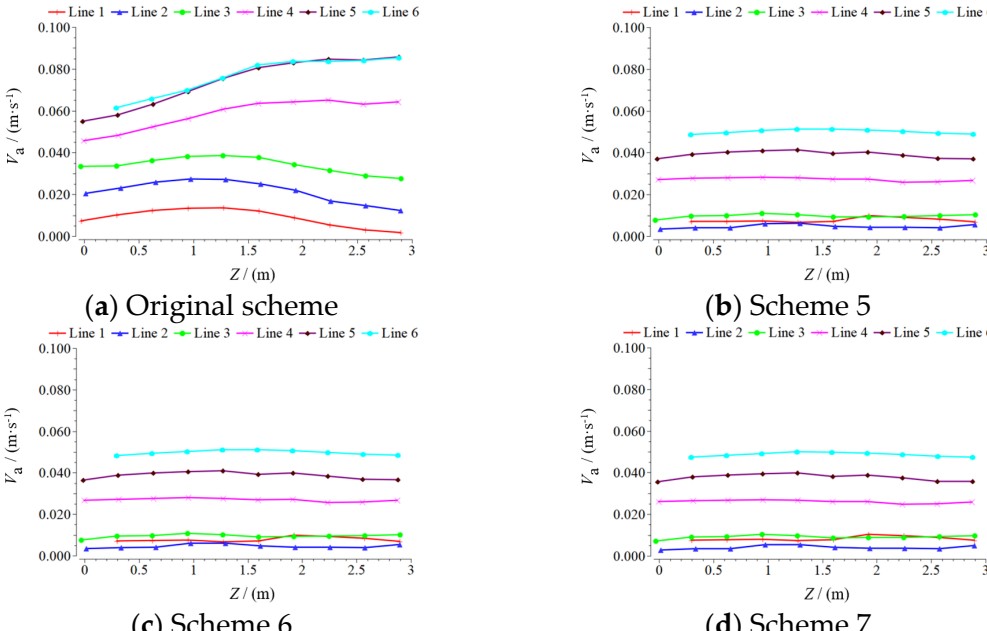

**Figure 15.** Diagram of axial velocity on the inlet of the sump 3#.

Figures 17–20 are the axial velocity chart before the bell mouth in the sump. In Figure 17a, the axial velocity decreases at a depth of 0.6 m, which is similar to the inlet of the sump. In Figure 17b,c, it is obvious that the axial velocity of each layer at the bell mouth is close to each other without obvious fluctuations by rectification. In Figures 18 and 19, the axial velocity starts to decrease at a depth of 1.5 m for each rectification scheme. The axial velocity has been significantly improved and the flow pattern has been optimized with the measures of three-sectional diversion pier, triangle column and straight back baffle, as shown in Figure 20.

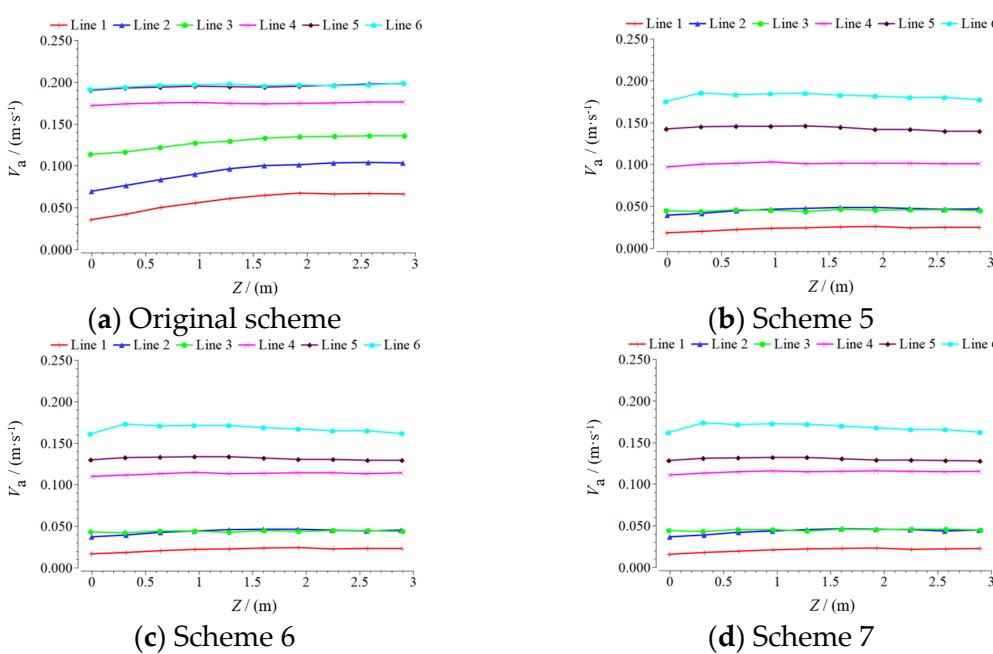

**Figure 16.** Diagram of axial velocity on the inlet of the sump 4#.

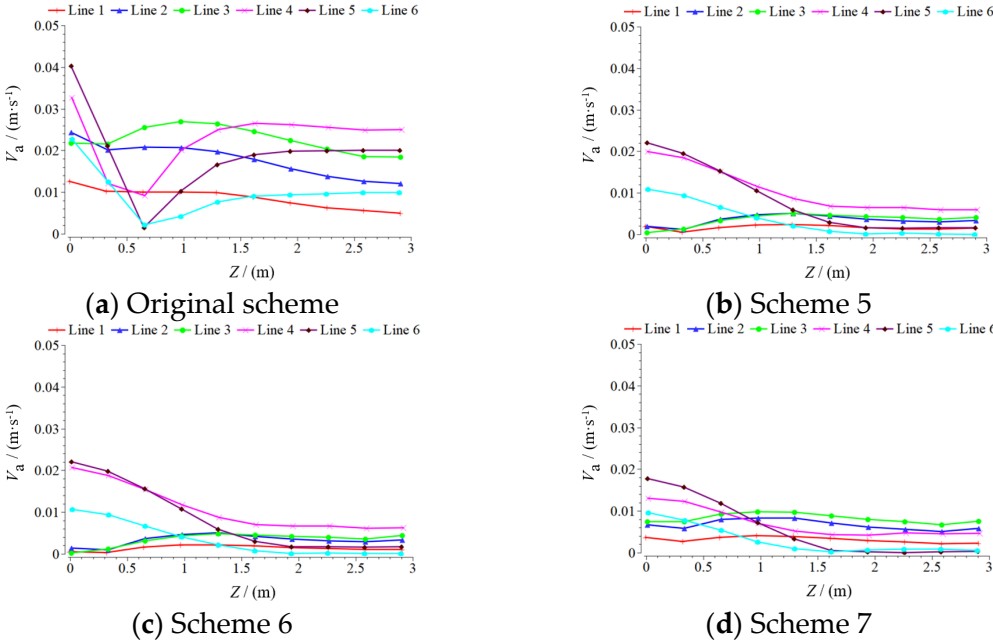

**Figure 17.** Diagram of axial velocity before the bell mouth in the sump 1#.

Table 4 shows the axial velocity uniformity and axial velocity weighted average angle of Section 5 in the sump for each scheme. As shown in the table, the axial velocity uniformity of the original scheme is poor. The maximum value is 56.94% and the minimum value is just 37.9%. Compared with the original scheme, axial velocity uniformity of Section 5 for schemes 5–7 in the four sumps are significantly improved, with an average increase of 34.9%. For scheme 6, the axial velocity uniformity in sump 1#–3# is excellent, and the smoothness of the flow in sump 4# does not change significantly. For scheme 7, the axial velocity uniformity of sump 1#–3# is slightly improved, and the axial velocity uniformity of sump 4# is slightly decreased. Moreover, the axial velocity weighted average angle of the original scheme is still relatively small. The maximum value is 61.17°, the minimum value is 33.49°. The axial velocity weighted average angle of schemes 5–7 in sump 1#–3#

is larger than 80°. The axial velocity weighted average angle of schemes 5–7 in sump 4# is more than 60°, which is almost two times than the original scheme.

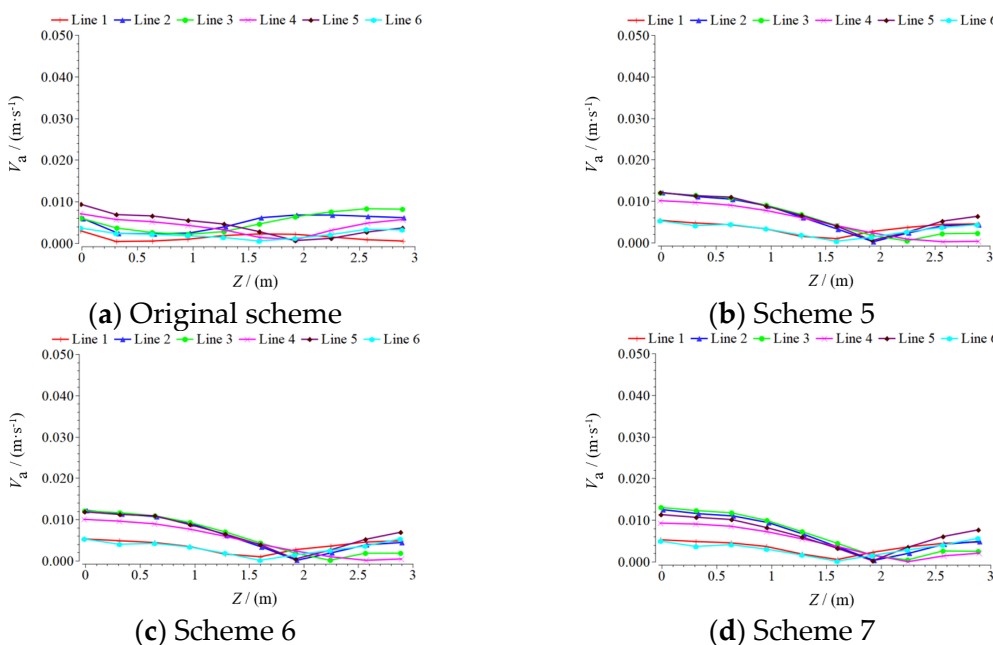

**Figure 18.** Diagram of axial velocity before the bell mouth in the sump 2#.

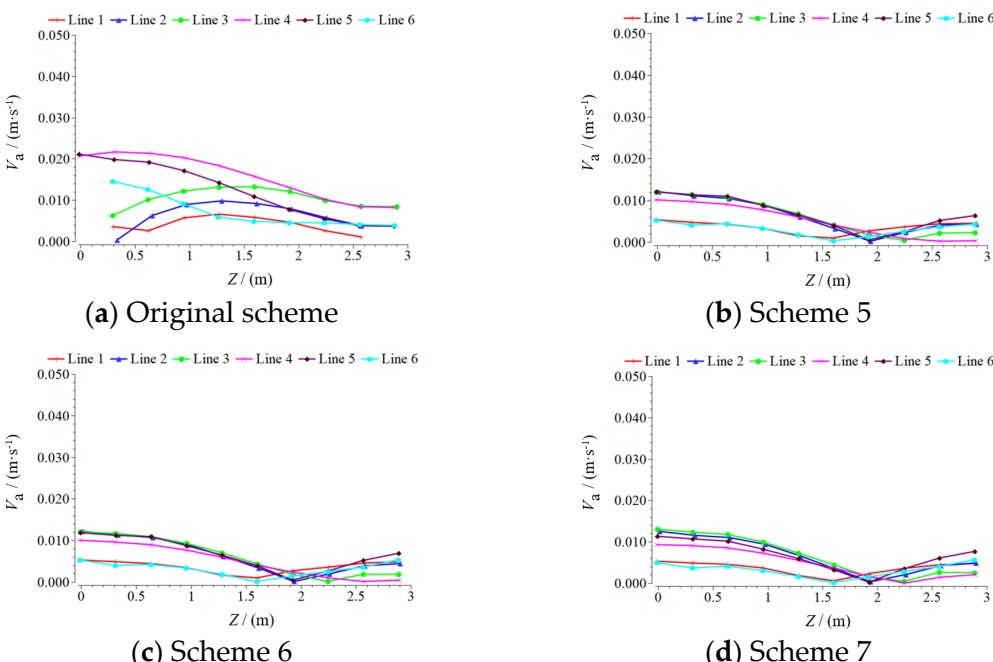

**Figure 19.** Diagram of axial velocity before the bell mouth in the sump 3#.

Table 5 is the axial velocity uniformity and axial velocity weighted average angle of Section 4 in the sump for each scheme. There is a large difference in the axial velocity uniformity of each sump for the original scheme. The maximum value is 76.51% and the minimum value is 16.74%. Compared with the original scheme, the axial velocity uniformity and smoothness in the sump 2#–4# of scheme 5 enlarges, increasing significantly by 36.5% on average. For schemes 6 and 7, the axial velocity uniformity of sump 1#–3# increases lightly. Nevertheless, the axial velocity uniformity of sump 4# further increases by 11.43% and 4.92%. The axial velocity weighted average angle of the original scheme is

79.85° at the maximum and 59.9° at the minimum. For the rectification schemes, the axial velocity weighted average angle is improved at more than 84°. Finally, scheme 7 is the advised scheme.

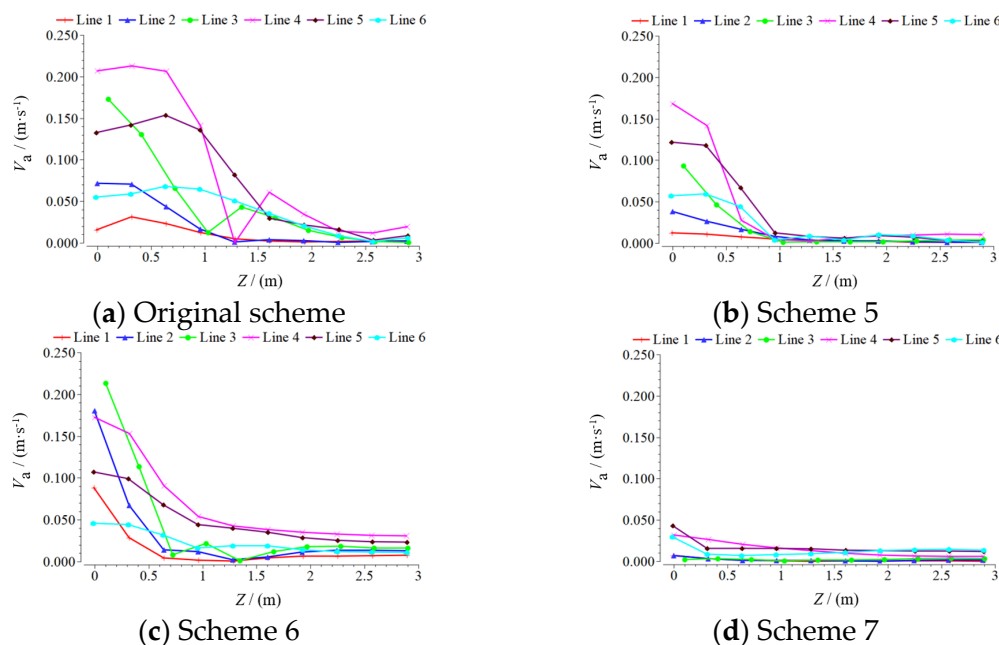

(**a**) Original scheme　　　　　　　　　　　　　　(**b**) Scheme 5

(**c**) Scheme 6　　　　　　　　　　　　　　　　(**d**) Scheme 7

**Figure 20.** Diagram of axial velocity before the bell mouth in the sump 4#.

**Table 4.** Axial velocity uniformity and axial velocity weighted average angle of Section 5 for each scheme.

| Scheme | 1# | | 2# | | 3# | | 4# | |
|---|---|---|---|---|---|---|---|---|
| | $V_{au}$/% | $\theta_a$/(°) | $V_{au}$/% | $\theta_a$/(°) | $V_{au}$/% | $\theta_a$/(°) | $V_{au}$/% | $\theta_a$/(°) |
| Original scheme | 37.89 | 61.17 | 56.94 | 42.34 | 47.05 | 36.90 | 48.29 | 33.49 |
| Scheme 6 | 81.61 | 86.51 | 84.31 | 83.60 | 81.87 | 80.67 | 81.88 | 60.66 |
| Scheme 7 | 81.61 | 86.51 | 84.33 | 83.61 | 81.86 | 80.75 | 81.98 | 60.88 |
| Scheme 8 | 81.64 | 86.48 | 84.31 | 83.61 | 81.95 | 80.93 | 81.68 | 61.15 |

**Table 5.** Axial velocity uniformity and axial velocity weighted average angle of Section 4 for each scheme.

| Scheme | 1# | | 2# | | 3# | | 4# | |
|---|---|---|---|---|---|---|---|---|
| | $V_{au}$/% | $\theta_a$/(°) | $V_{au}$/% | $\theta_a$/(°) | $V_{au}$/% | $\theta_a$/(°) | $V_{au}$/% | $\theta_a$/(°) |
| Original scheme | 76.51 | 59.90 | 40.83 | 74.20 | 42.12 | 73.89 | 16.74 | 79.85 |
| Scheme 5 | 80.12 | 84.93 | 79.56 | 84.41 | 75.33 | 85.06 | 54.44 | 84.26 |
| Scheme 6 | 80.32 | 85.01 | 79.62 | 84.58 | 75.72 | 85.56 | 65.87 | 84.72 |
| Scheme 7 | 80.38 | 85.01 | 79.73 | 84.71 | 75.75 | 85.56 | 70.79 | 84.82 |

#### 5.2.2. Outflow Discharge Balance

Figure 21 shows the outflow discharge of each suction pipe for each scheme. For original scheme, the outflow discharge of each suction pipe differs too much. Discharge D-value between the outlet of the suction pipe 1# and 4# is almost 70 kg·s$^{-1}$. When the rectification measures are installed, the difference in the outlet of the suction pipe is narrowed, the outflow discharge becomes more balanced, especially for scheme 8. The standard deviation and variance were used to assess the degree of dispersion of a data set. The smaller the value is, the closer to the average is. In this manuscript, this can reflect on the stability

of the outflow discharge to a certain extent. For the original scheme, the standard deviation and variance are 751.06 and 27.41. For scheme 5, the standard deviation and variance are 100.39 and 10.02. For scheme 6, the standard deviation and variance are 189.08 and 13.75. For scheme 7, the standard deviation and variance are 3.32 and 1.82. As a result, it is further verified that the flow is basically stable by the combined rectification measures of "three-sectional diversion pier + column + straight back baffle" in sumps.

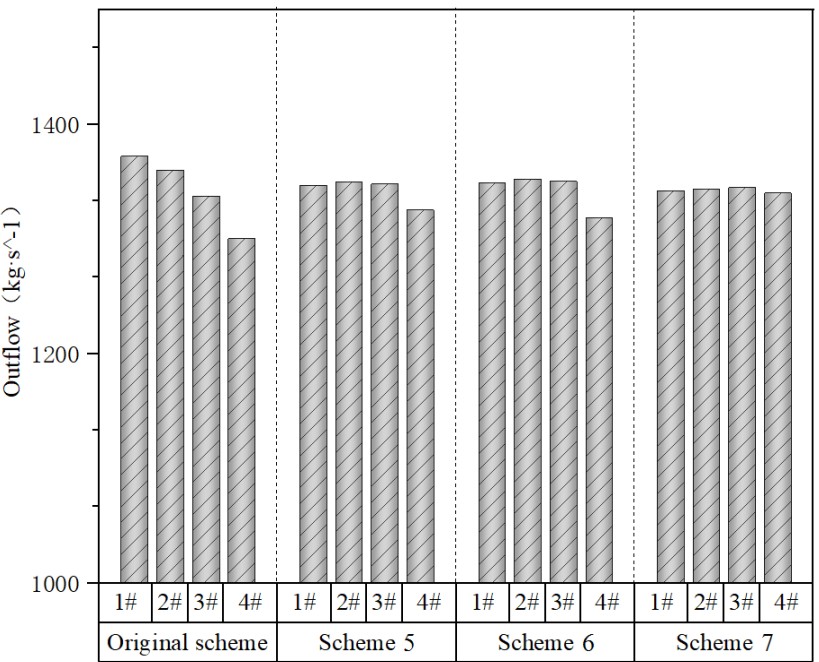

**Figure 21.** Outflow discharge of each scheme.

## 6. Conclusions

To solve the turbulent flow in a pump station with laterally asymmetric inflow, a recent flow pattern was numerically simulated and analyzed, and then the effective vortex elimination measures such as the "straight diversion pier + curved wing wall", "straight diversion pier + curved wing wall + V-shaped diversion pier", " symmetrical 川-shaped diversion pier", " symmetrical 川-shaped diversion pier + circular column" "three- sectional diversion pier", "three- sectional diversion pier + triangle column" and "three-sectional diversion pier + triangle column + straight back baffle" were proposed. The following conclusions can be drawn:

(1) For the original scheme, the flow pattern in the forebay was extremely irregular. A large-scale recirculation zone was observed, covering about 40% region of the forebay. The existing recirculation zone pushed and squeezed the mainstream in the intake channel. Flow separation and obvious vortices were observed;

(2) For each anti-vortex measures of each scheme in the forebay, the flow pattern was improved. Among the four schemes, the combined anti-vortex measures of "straight diversion pier + curved wing wall + V-shaped diversion pier + symmetrical 川-shaped diversion pier + circular column" effectively improved the flow pattern in the forebay. The clash inflow in the south and north intake channel was suppressed by applying the "straight diversion pier + curved wing wall + V-shaped diversion pier". The "symmetrical 川-shaped diversion pier" and "symmetrical 川-shaped diversion pier + circular column" were adopted to weaken the bias flow and large-scale recirculation zone in the forebay. The volume of the recirculation and the vortices were reduced by 44%;

(3) All the rectification schemes can constrict the vortices and asymmetric flow near the suction pipe in the sump. However, the combined rectification measures of "three-sectional diversion pier + triangle column + straight back baffle" are the most effective.

Axial velocity uniformity and axial velocity weighted average angle greatly increased, especially on the section before the bell mouth of the suction pipe, with values enhanced by an average of 32.61% and 13.07°, respectively.

**Author Contributions:** Data curation, C.L. and L.C.; Methodology, C.L. and X.C.; Formal analysis, Y.S.; Writing—original draft, C.L., Y.H. and S.L.; Writing—review and editing, C.L., Y.H. and C.D.; Supervision, C.D. All authors have read and agreed to the published version of the manuscript.

**Funding:** This research was funded by Jiangsu Province Science Foundation for Youths (Grant no. BK20170507), Postgraduate Research & Practice Innovation Program of Jiangsu Province (Grant no. SJCX22_1759), National Natural Science Foundation of China (Grant No. 52279091, 522019116), Natural Science Foundation of the Jiangsu Higher Education Institutions (Grant no. 17KJD580003), Jiangsu Planned Projects for Postdoctoral Research Funds (Grant no. 1701189B), Priority Academic Program Development of Jiangsu Higher Education Institutions (PAPD).

**Institutional Review Board Statement:** Not applicable.

**Informed Consent Statement:** Not applicable.

**Data Availability Statement:** Not applicable.

**Acknowledgments:** The authors thank the College of Hydraulic Science and Engineering, Yangzhou University. The author is very grateful for the discussions with Luo C and Lei S H. A huge thanks is due to the editor and reviewers for their valuable comments to improve the quality of this paper.

**Conflicts of Interest:** The authors declare no conflict of interest.

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
