# Peer review of "Flow Characteristics and Anti-Vortex in a Pump Station with Laterally Asymmetric Inflow"

_processes, doi:10.3390/pr10112398_

Round 1

Reviewer 1 Report

At the beginning, the article outlines the issue of flow in a pumping station, where, due to the inflow of liquid to the suction pipe through various inlet parts (pre-bar, sump), excessive swirling occurs and thus hydraulic losses. Design modifications without the possibility of experimentation are very limited. In this case, mathematical flow modeling is a very suitable tool for verifying flow trends. At the same time, it is possible to test a number of variants. In the thesis, the methodology for the design of new geometries of the entrance part of the pumping station is very clearly presented, a comparison system is defined using suitable parameters for evaluating the properties of individual variants. The numerical solution is very clearly displayed graphically.

The chapters of the article follow each other logically, the graphic processing is at a high level. Unfortunately, the results are not compared with the experiment, which is desirable in a numerical solution approach. However, a trend can be traced in the evaluation of the examined flow parameters, and the conclusion of the work is focused on the evaluation of the proposed variants of the construction of the inlet spaces to the pump.

The work uses high-quality mathematical approaches and the results can benefit the field of structural design. The methodology is at an excellent level, the article is an asset in the field of applied sciences.

After accepting my comments, I recommend the article for publication.

Comments:

The abstract is too comprehensive and contains detailed conclusions. Prepare a brief abstract with general solution methods and give the specific numerical evaluation of the method only in the conclusion.

In the introduction, insert a schematic figure  clarifying the problem being solved.

In flow theory, the term "Vortices" is used, not "Vortexes". Replace with text.

Correct the denominator in equation (1)

Correct in equation (3) uj, j is the index. Viscosity does not have indices ij, but t as turbulent viscosity (according to Boussinesq theory)

Line 128 – correct the sentense : The inlet of the intake channel set sets

Line 131 - why the water level is defined as symmetry? It doesn't make sense physically

Line 135-136 – add whether the problem is solved as a time-dependent problem and with what time step or if it is solved as a steady problem. Complete other parameters of the numerical model, e.g. differential scheme for pressure, solution method (SIMPLE or other), numerical method used for solution (method of finite volume), or others.

Figure 9 - meshes are unreadable, attach mesh details mainly to transition from one mesh type to another

Line 153 - define h

Equation (7) is the general definition of the characteristic equation. It is not used in further text, nor is its meaning for further use explained. Specify the meaning of equation (7) and its coefficients or do not specify it.

Table 3 - correct the name of the first column "Scheme"

Figure 7 – the description and labeling of the figure must be on the same page as the figure

Line 338 - delete the last word "it", the sentence does not make sense

Reviewer 2 Report

This paper deals with a very interesting and important topic - the influence of different anti-vortex devices on flow in a laterally asymmetric pump station intake. These are some remarks and recommendations:

 1.         The paper requires moderate English grammar corrections. Also, some strange formulations should be rephrased.

2.         Abstract is too long and should be shortened.

3.         Concerning References, the reference number should follow just the authors’ names. References [23] and [24] are missing. References [22] and [25] are written as the upper indexes. There is no citation on the NX 12.0 software (theory or user) manuals.

4.         Equation (1) uses variables x, u as the lower indexes. The dynamic viscosity in Equation (3) is a scalar and should not use indexes.

5.         Equations (1)-(3): Authors describe just the laminar flow, but they model the turbulent flow. They should at least mention the turbulence model characteristics and the appropriate reference.

6.         Section 2.3: Where are the pumps? Do authors consider at least the pre-rotation generated by pumps in the sump?

7.         Line 154, Figure 3: How is the hydraulic loss evaluated? Is it related to the original scheme?

8.         Lines 158-159: Is the mesh with 1.2 mil. grid nodes related to the original scheme?

In any case, the number is very small to capture complex vortical structures inside the pump station intake; especially for the schemes 4, 6, 7 with many piers and columns. What is the size of control volumes in the mesh core and close to columns? Authors should discuss and give pictures of the mesh in the critical regions.

9.         Section 3.2: Authors should correctly define the axial and tangential velocity and related coordinate system.

10.       Lines 219-228: Numbering of the schemes does not correspond to Table 3.

11.       Table 3 top-left cell: Scheme

12.       Table 4: Authors should give a detail of the triangle column geometry.

13.       In the paper, there is just comparison of vortical structures inside different schemes of the pump station intake. Authors should add data on hydraulic losses of different schemes, as these are highly important for the pump station operation.

14.       Authors use just the symmetry plane instead of the free surface modelling. Consequently, they are not able to evaluate water level drop in front of the pump bell mouth, which is important for the pump suction characteristics. They should at least discuss.

15.       I like using Chinese character chuān/creek to describe the shape of diversion piers. It is fresh and fitting :-).

Reviewer 3 Report

The paper presents a high scientific level. Results presented clearly. Minor comment in the manuscript commentary.
